# The Effect of Intravenous Tranexamic Acid on Perioperative Blood Loss, Transfusion Requirements, Verticalization, and Ambulation in Total Knee Arthroplasty: A Randomized Double-Blind Study

**DOI:** 10.3390/medicina60071183

**Published:** 2024-07-21

**Authors:** Gordana Jovanovic, Mirka Lukic-Sarkanovic, Filip Lazetic, Teodora Tubic, Dajana Lendak, Arsen Uvelin

**Affiliations:** 1Faculty of Medicine, University of Novi Sad, 21000 Novi Sad, Serbia; mirka.lukic-sarkanovic@mf.uns.ac.rs (M.L.-S.); teodora.tubic@mf.uns.ac.rs (T.T.); dajana.lendak@mf.uns.ac.rs (D.L.); arsen.uvelin@mf.uns.ac.rs (A.U.); 2Clinic for Anesthesia, Intensive Care and Pain Therapy, University Clinical Center of Vojvodina, 21000 Novi Sad, Serbia; 3Clinic for Orthopedic Surgery and Traumatology, University Clinical Center of Vojvodina, 21000 Novi Sad, Serbia; lazetic90@gmail.com; 4Clinic for Infectious Disease, University Clinical Center of Vojvodina, 21000 Novi Sad, Serbia

**Keywords:** tranexamic acid, total knee arthroplasty, perioperative bleeding, transfusion, verticalization, ambulation

## Abstract

*Background and Objectives:* Total knee arthroplasty (TKA) is sometimes associated with significant perioperative bleeding. The aim of this study was to determine the efficacy of tranexamic acid (TXA) in reducing perioperative blood loss in patients undergoing primary TKA. The secondary objectives were to assess the efficacy of TXA in reducing the need for blood transfusion in these patients and to determine its effect on verticalization and ambulation after TKA. *Materials and Methods:* This study included 96 patients who were randomly assigned to two groups, each containing 48 patients. The study group received intravenous TXA at two time points: immediately after the induction with doses of 15 mg/kg and 10 mg/kg 15 min before the release of the pneumatic tourniquet. The control group received an equivalent volume of 0.9% saline solution via the same route. *Results:* TXA markedly reduced (Z = −6.512, *p* < 0.001) the total perioperative blood loss from 892.56 ± 324.46 mL, median 800 mL, interquartile range (IQR) 530 mL in the control group, to 411.96 ± 172.74 mL, median 375 mL, IQR 200 mL, in the TXA group. In the TXA group, only 5 (10.4%) patients received a transfusion, while in the control group, 22 (45.83%) received it (χ2 = 15.536, *p* = 0.001). Patients in the study group stood (χ2 = 21.162, *p* < 0.001) and ambulated earlier postoperatively, compared to the control group (χ2 = 26.274, *p* < 0.001). Patients who received TXA had a better overall postoperative functional recovery. There was a statistically significant difference in all the above results. *Conclusions:* TXA is an effective drug for reducing the incidence of perioperative bleeding, decreasing transfusion rates, and indirectly improving postoperative functional recovery in patients undergoing primary TKA.

## 1. Introduction

Although discovered more than 60 years ago, there is perpetual interest in tranexamic acid (TXA) and its use in medicine and surgery. Today, it has been widely used in cardiac surgery, orthopedic surgery, and gynecologic surgery, followed by its most recent use in patients with trauma and neurosurgery [1,2].

Advancements in surgical and anesthesia techniques have made the implantation of total knee prostheses, referred to herein as total knee arthroplasty (TKA), a common and safe procedure. However, being an extensive orthopedic intervention, it is sometimes associated with significant blood volume loss [3,4].

Midway through the 20th century, Finnish authors led by Hippala carried out the first studies on the use of tranexamic acid in TKA [5,6]. Afterwards, several studies have been conducted, particularly in the past 20 years, using various administration routes (intravenous, intra-articular-topical, periarticular, and oral), various dosages (high > 20 mg/kg, or low dosage < 20 mg/kg), dosing regimens (single dose, multiple doses), and combinations of all these approaches [7,8]. Through these studies, TXA consistently demonstrated the ability to reduce blood loss during primary TKA. The first meta-analyses showed that use of TXA reduced total blood loss by a mean of 487 mL, intraoperative blood loss by a mean of 127 mL, and postoperative blood loss by a mean of 406.69 mL [4]. Latter studies proved the safety of the TXA, as there were no significant differences in deep-vein thrombosis (DVT), pulmonary embolism, or other complications among the patients who received the TXA [9,10].

The American Association of Hip and Knee Surgeons, American Society of Regional Anesthesia and Pain Medicine, American Academy of Orthopaedic Surgeons, Hip Society, and Knee Society, in their recent guidelines [11], strongly recommend using TXA in primary total joint replacement (TJA) surgery. These recommendations are frequently followed, and TXA is widely used in TJA surgery; however, not all hospitals have adopted them as standard practice just yet [11].

There are three possible directions for future TXA research: use in high-risk patients (with a history of deep venous thrombosis, myocardial infarction, stent placement, or cerebrovascular insult), research with different routes of administration and their combinations, and research of TXA as a part of enhanced recovery programs [12,13]. This study addresses certain gaps in the existing literature, such as providing further evidence of specific dosing and the lack of studies that have examined the secondary outcomes of functional recovery after TKA. 

Our study was part of a larger institutional attempt to advance perioperative patient blood management (PBM) care in TKA. Our goal was to examine the specific effects of tranexamic acid in orthopedic surgery as a component of a larger, more comprehensive recovery regimen [14,15].

Intraoperative blood loss and bleeding in the joint cavity and surrounding periaticular tissue can cause swelling, pain, and delayed function. Allogenic transfusion is a lifesaving procedure, but not without complications [16]. Allogenic blood is a spare and very valuable biological product, and blood products are extensively used in orthopedic surgery (up to 10% of all hospital consumption). Hospitals are making major attempts to promote and develop procedures and clinical care pathways that can have a blood-sparing effect [17]. Tranexamic acid, as a safe, simple, and cost-effective drug, can be a valuable component of such clinical care pathways.

### 1.1. Hypothesis

The null hypothesis was that there is no difference between the two groups: the TXA group receiving double intravenous doses of tranexamic acid vs. the control group receiving double doses of placebo, in perioperative blood loss, transfusion rate, and postoperative verticalization and ambulation.

Our hypothesis was that there is a statistically significant difference between the two groups (TXA vs. control) in perioperative blood loss, transfusion rate, and postoperative verticalization and ambulation.

### 1.2. Objectives

Primary objective: this trial was designed to determine the efficacy of TXA in reducing perioperative blood loss in patients undergoing primary TKA. Secondary objectives: to determine the efficacy of TXA in reducing the need for blood transfusion in patients undergoing primary TKA and to assess its effect on verticalization and ambulation after TKA.

## 2. Materials and Methods

This research was conducted as a single-centered, controlled, prospective, randomized, double-blind study at the Clinic for Orthopedic Surgery and Traumatology of the University Clinical Center of Vojvodina in Novi Sad. This study was approved by the Ethics Committee of the Clinical Center of Vojvodina in Novi Sad (Serbia). Document no. 00-03/97. This study was conducted in accordance with the principles of the Declaration of Helsinki. This study took place from October 2012 to December 2014.

### 2.1. Patients

#### 2.1.1. Inclusion Criteria

The inclusion criteria were adult female and male patients who underwent primary, unilateral TKA due to degenerative knee diseases, with ASA status 1–3, and provided written informed consent to participate in this study.

#### 2.1.2. Exclusion Criteria

Patients with a known allergy to TXA, with Hgb less than 13 g/dL for men and 12 g/dL for women, coagulation disorders, a previous history of any type of thromboembolic event, or those who received fresh frozen plasma, other blood products, or drugs affecting the coagulation system 24 h before surgery were excluded from this study. Additionally, patients with severe heart disease (New York Heart Association Classification-NYHA III and IV), creatinine values above 115 μmol/L for men and 100 μmol/L for women, elevated liver enzymes, or congenital thrombophilia were also excluded.

#### 2.1.3. Sample Size Calculation

Sample size calculated for primary outcome (perioperative bleeding), for effect size 0.5, with significance level α = 0.005, and P (power) of 90% (0.9) is 43 in each group. 

### 2.2. Anesthesia and Surgery 

All patients received spinal anesthesia (level L3 and L4 lumbar spine) with isobaric bupivacaine 0.5% (Marcaine^®^ spinal 0.5%, Astra Zeneca, Cambridge, UK) at a dose of 15 mg (3 mL) (Figure 1). Standard intraoperative monitoring was conducted, including continuous monitoring of heart function by electrocardiogram (D II lead), non-invasive blood pressure measurement, and pulse oximetry (Infinity Delta XL, Drager, Lubeck, Germany). The night before the surgery, all participants received subcutaneous low molecular weight heparin nadroparin calcium (Fraxiparine 2850 IU/0.3 mL, Glaxo Smith Kline, Middlesex, UK), dosed according to body weight.

All TKAs were performed by the same surgical team, using either the anterior parapatelar sub-vastus or, in cases of reduced knee motility, the mid-vastus approach. During surgical field preparation, the leg was elevated (for 5 min, 45°) to drain blood vessels. All surgeries in this study were performed with the placement of an uninflated pneumatic tourniquet at the proximal thigh before the start of surgical field preparation. The tourniquet was inflated with the knee in flexion prior to the incision. Safe tourniquet pressure was calculated as a fixed amount of pressure above systolic arterial pressure (100–150 mmHg above systolic arterial pressure, measured at the beginning of the procedure, before the application of spinal anesthesia). The pneumatic tourniquet was deflated after the cementing of the prosthesis (from skin incision to cement hardening). An intra-articular drainage tube (Polymed^®^ Medical Devices, Faridabad, India) was inserted at the conclusion of the procedure and removed 24 h later. After the procedure, a modified Robert-Jones elastic bandage was placed.

### 2.3. Method of Drug Administration

The study group received TXA (Tranexamic Acid^®^ Medochemie 500 mg/5 mL, Medochemie Ltd., Limassol, Cyprus) at two time points (T1 and T2), with doses of 15 mg/kg and 10 mg/kg, respectively. The drug was administered as a continuous intravenous (i.v.) infusion over 15 min. The first time point (T1) was immediately after the induction of anesthesia, while the second time point (T2) was 15 min before the release of the pneumatic tourniquet (Figure 1).

The control group received the same volume of 0.9% saline solution via the same route (i.v.) (Figure 1).

The attending anesthesiologist and anesthesia assistant were blinded to IV infusion content. Patients were allocated using a random selection method.

### 2.4. Blood Loss Measurement

During this research, blood loss was measured as intraoperative and postoperative blood loss. The intraoperative loss was estimated by visually examining graduated suction canisters and drains expressed in milliliters, and surgical gauze or sponge loss was determined using the gravimetric method.

This method involved measuring dry surgical sponges (15 × 6 cm). At the end of the operation, all the blood-soaked surgical sponges were counted and weighed again. Estimated blood loss is determined by assessing the weight difference before and after their use, with every gram of weight equivalent to 1 mL of blood loss.

The postoperative blood loss was measured after 6, 12, and 24 h, including drain loss measured in milliliters and total postoperative loss from 0 to 24 h.

### 2.5. Transfusion Rate

The transfusion rate was calculated as the total number of allogenic blood units given intraoperatively or up to 48 h postoperatively.

The first hemoglobin levels were measured immediately after the operation, on the zero postoperative day, and on the second postoperative day. The hemoglobin level at which allogeneic blood transfusions were initiated was 9 g/dL, according to the local institutional protocol.

### 2.6. Crystalloid Fluid Administration 

Intraoperatively administered crystalloid solutions were measured both as the total amount per patient in milliliters and as the average amount per patient in milliliters.

### 2.7. Verticalization and Ambulation 

In the postoperative period, patients were transferred from the postanesthesia care room to the surgical ward, and they started oral ingestion of fluids as soon as the residual effects of spinal anesthesia were totally gone. Intravenous infusions were given only for drug administration. A zero-postoperative day was defined as the day surgery was performed.

Every patient received the same multimodal analgesia, according to our institutional protocol, including a weak opioid (tramadol, Trodon^®^ Hemofarm AD Vršac, Serbia, 100 mg/2 mL) only on the first postoperative day and nonsteroidal anti-inflammatory drugs (ketorolac, Zodol^®^ 30 mg/mL, Hemofarm AD Vršac, Serbia, and paracetamol, Paracetamol^®^ Actavis d.o.o. Belgrade, Serbia, 10 mg/mL), from the first postoperative day until needed.

Following surgery, each patient underwent the same early rehabilitation regimen led by a ward physiotherapist. On day zero, active knee flexion was started, led by a physiotherapist. Passive knee flexion was not used.

Successful verticalization and ambulation were defined as the first time to verticalization (standing up with the walking frame) and the first time to ambulation (walking with the walking frame). After successful walking with the walking frame, the next ambulation was planned with a walking stick.

Only clinical signs of potential thromboembolic complications were noted. 

### 2.8. Post-Discharge Follow-Up

All patients received low-molecular-weight heparin up to stitch removal.

Every patient had an ambulatory visit between the fifteenth and nineteenth days following surgery. After examining the patient, the operating surgeon removed the stitches.

### 2.9. Statistical Analysis

Data analysis was performed using SPSS 26 for Windows.

Standard methods of statistical research, including descriptive statistics and frequency distribution, were used in the analysis. Numerical data were presented through mean arithmetic values, standard deviation, median and interquartile range (IQR), frequencies, and percents. The threshold values for outliers were 1.5 × IQR and for extremes, 3 × IQR above or below the third quartile. Comparisons of the examined groups were performed using the *t*-test and the Mann–Whitney U test. Pearson’s χ2 test was used to test the difference in frequency (distribution) between the two groups. The normality of the data was assessed using skewness, kurtosis, Kolmogorov–Smirnov tests, and Shapiro–Wilik tests.

To analyze the differences in the amount of blood loss within a group after 6, 12, and 24 h, the Friedman test for repeated measurements was used. To examine where the differences actually occur, a Wilcoxon signed-rank test on the different combinations of related groups was used. In order to avoid Type I error, a Bonferroni adjustment was used (the initially used significant level of 0.05 was divided by the number of tests by 3), and *p* < 0.017 was considered a statistically significant result.

Binary logistic regression was performed to find independent predictors of the probability that an allogeneic blood transfusion would be administered. Data were presented as an OR-odds ratio and a 95% CI—confidence interval. The significance level was set at a *p*-value less than 0.05.

## 3. Results

### 3.1. Patient Characteristics 

A total of 96 patients were included in this study and divided into the study group (48 patients) and the control group (48 patients). The study group received TXA intraoperatively, while the control group received a 0.9% saline solution (placebo).

The analysis of the basic demographic characteristics of the groups, as well as the preoperative health status of the patients, indicates their uniformity and excludes the influence of these parameters on further research (Table 1).

### 3.2. Intraoperative Blood Loss

The average intraoperative blood loss in the study group was 102.17 ± 93.52 mL, and in the control group, it was 438.84 ± 287.68 mL (Appendix A).

There is an asymmetrical distribution of intraoperative blood loss values in both observed groups.

The median in the TXA group was 100 mL (95% CI 50.00–142.98 mL), and the interquartile range was 162.5 mL (0–162.5 mL). One quarter of patients (up to 25%) in the TXA group had no bleeding; a quarter of patients (out of 75%) in the TXA group had intraoperative bleeding over 162.5 mL. One patient (outlier patient no. 48) had a blood loss of 400 mL (Figure 2).

In the control group, the median was 400 mL (95% CI 242.47–500.00 mL), and the interquartile range was 350 mL (100–450 mL). One-quarter of patients had blood loss up to 100 mL, and one-quarter of patients had blood loss over 450 mL. One patient (outlier patient no. 61) had a blood loss of 1400 mL (Figure 2).

The results of the Mann–Whitney U test show statistically significantly higher intraoperative blood loss in the control group compared to the study group (Z = −6.931; *p* < 0.001) (Appendix A).

### 3.3. Postoperative Blood Loss

#### 3.3.1. Longitudinal Analysis within the Groups

When observing the trend of bleeding within the groups, the average postoperative blood loss in the TXA group was highest after 6 h (123.91 ± 75.82 mL), lowest after 12 h (90.22 ± 82.74 mL), and 95.65 ± 97.08 mL after 24 h (Appendix A).

The Friedman test determined that there was no statistically significant difference in blood loss values at 6, 12, and 24 h measurement times in the TXA group (χ2 = 5.82, *p* = 0.055).

In the control group, the highest average postoperative blood loss was after 6 h, 203.72 ± 100.12 mL, 132.40 ± 114.97 mL after 12 h, and the lowest after 24 h, 113.95 ± 86.14 mL. 

The Friedman test revealed a statistically significant variation in blood loss in the control group in repeated measurements (6 h, 12 h, 24 h) (β2 = 14.90, *p* = 0.001). Post hoc analysis with Wilcoxon signed-rank tests was conducted with a Bonferroni correction applied. A statistically significant difference in the amount of blood loss was obtained between the values obtained at 6 h and 12 h (Z = −3.567, *p* < 0.001) and between 6 h and 24 h (Z = −3.194, *p* < 0.001).

#### 3.3.2. Comparison of Postoperative Blood Loss between the Groups

The Mann–Whitney test determined that there was a statistically significant difference between the TXA and control group in the measured values of blood loss only after 6 h (Z = −4.511, *p* < 0.001), while no difference was found in the values after 12 h and 24 h (*p* > 0.05).

The total average postoperative blood loss was 309.78 ± 143.61 mL in the TXA group and 449.07 ± 196.31 mL in the control group (Appendix A).

Patients in the control group had statistically significantly higher total postoperative blood loss than patients in the study group (Z = −4.319, *p* < 0.001).

The median postoperative blood loss after 6 h, 12 h, and 24 h was 100 mL (95% CI 100.00–142.98 mL, 50.00–100.00 mL, and 32.02–100.00 mL). The interquartile range after 6 h, 12 h, and 24 h was 50, 162.5, and 150 mL. After 6 h, half of the patients had a blood loss of 50 to 100 mL. One patient had a blood loss of 500 mL (extreme). After 12 h, half of the patients had blood loss up to 162.5 mL, and after 24 h, half of the patients had blood loss up to 100 mL. After 24 h, one patient had a blood loss of 400 mL (outlier) (Figure 3).

In the control group after 6 h, the median was 200 mL (95% CI 150.00–200.00 mL), after 12 h, 100 mL (95% CI 64.04–185.95 mL), and after 24 h, 100 mL (95% CI 50.00–100.00 mL). The interquartile range after 6 h, 12 h, and 24 h was 150 mL. After 6 h, half of the patients had blood loss of 100 to 250 mL. One patient had a blood loss of 600 mL (outlier). After 12 h and 24 h, half of the patients had blood loss up to 150 mL. After 12 h, one patient had a blood loss of 550 mL (outlier) (Figure 3).

In the TXA group, the median for total postoperative blood loss was 300 mL (95% CI 250.00–350.00 mL), and the IQR was 125 mL. Half of the patients in the TXA group had blood loss of 100–175 mL. One patient had a blood loss of 850 mL (outlier) (Figure 3).

In the control group, the median for total postoperative blood loss was 450 mL (95% CI 350.00–485.95 mL), and the IQR was 150 mL. Half of the patients in the control group had blood loss of 100–250 mL. One patient had a blood loss of 1350 mL (asterisk) (Figure 3).

### 3.4. Perioperative Blood Loss

There was a statistically significant difference (Z = −6.512, *p* < 0.001) in total perioperative blood loss between the two groups. TXA markedly reduced total average perioperative blood loss from 892.56 ± 324.46 mL in the control group to 411.96 ± 172.74 mL in the TXA group. In the TXA group, the median was 375 mL, IQR 200 mL (95% CI 300.00–450.00 mL), and in the control group, the median was 800 mL, IQR 530 mL (95% CI 714.04–985.95 mL). Maximal perioperative blood loss in the TXA group was 850 mL, and minimal perioperative blood loss was 150 mL. In the control group, maximal perioperative blood loss was 1700 mL, and minimal perioperative blood loss was 450 mL (Appendix A).

### 3.5. Transfusion Rate

In total, 32 units of allogeneic blood were transfused, with 5 units in the study group and 27 units in the control group. This difference is statistically significant (Z = −3.906, *p* < 0.01).

In the study group, 5 (10.42%) subjects received a transfusion, while in the control group, 22 (45.83%) received it. The need for transfusion was statistically lower in the study group (χ2 =15.536, *p* = 0.001) (Table 2).

Binary logistic regression analysis showed that preoperative hemoglobin, intraoperative bleeding, and being in a control group were the strongest predictors of the probability that an allogeneic blood transfusion would be administered. Age, gender, and obesity had no statistically significant effect (Table 3).

### 3.6. Intraoperative Crystalloid Solutions 

Intraoperatively, the average amount of administered crystalloid solutions (Ringer lactate) per patient was 1844.68 ± 415.88 mL in the TXA group and 1597.92 ± 346.10 mL in the control group, reaching a statistically significant difference in the TXA group (*t* test, t = 3.146, *p* < 0.01).

### 3.7. Postoperative Recovery

The length of hospitalization did not differ remarkably between the groups. In the study group, hospitalization ranged from 4 to 10 days, and in the control group, from 3 to 15 days (Z = −0.570, *p* = 0.568).

In the TXA group, 83.35% of patients had their first meal on postoperative day zero, while in the control group, 60.45% of patients had their first meal on day zero. These differences were statistically significant (χ2 = 5.942, *p* = 0.022). In this study, 28 (58.3%) patients from the TXA group were standing upright (verticalization) on day zero postoperatively, as opposed to 6 patients from the control group (12.5%). The difference was statistically significant (χ2 = 21.162, *p* < 0.001).

Half of the patients n = 24 (50.00%) in the study group were ambulating on postoperative day zero, while two patients n = 2 (4.20%) were ambulating in the control group. The difference was statistically significant (χ2 = 26.274, *p* < 0.001) (Appendix A).

## 4. Discussion

In line with recent research, our study showed that intravenous TXA administered in two intraoperative doses effectively lowers intraoperative, postoperative, and perioperative blood loss in primary TKA [4,8,10].

### 4.1. Primary Outcomes

#### 4.1.1. Intraoperative Blood Loss

One of the main findings of our research was the efficacy of TXA in reducing intraoperative blood loss. There was a statistically significant difference in intraoperative bleeding between the groups. Intraoperative bleeding in the control group was almost four times higher than in the study group receiving TXA.

In the literature, there are scarce data on intraoperative bleeding in TKA with the use of TXA. Early works reported data about intraoperative bleeding [5,6], but the majority of recent meta-analyses did not report intraoperative bleeding [8,10,13,18,19]. Intraoperative bleeding was recorded only in one meta-analysis in four RCTs [4], and there were no group differences. The majority of RCTs’ intraoperative bleeding was treated as clinically insignificant, which was explained by the action of the pneumatic tourniquet.

PT itself can have an impact on hemostasis and intraoperative bleeding. When inflated, there is catecholamine release as a response to physical pressure on the limb, which can lead to a hypercoagulable state. On the other hand, after releasing the pressure (deflation), there is a state of hyperfibrinolysis, which can cause increased intraoperative bleeding. This effect is maximal about 15 min after release, can last up to 12 h, and can be prolonged into the postoperative period [20,21]. The same logic of this pathophysiological effect led Hippala and coauthors to use tranexamic acid for the first time in their groundbreaking study [5,6].

In our research, pneumatic tourniquets were used for everything from skin incisions to cement hardening. There was no statistical difference in the duration of PT time between the groups or in the duration of the operation time. All intraoperative bleeding occurred during the time frame from deflating the PT to wound closure. We observed statistically significant intraoperative bleeding in the control group and clinically negligible bleeding in the TXA group. In their meta-analysis, Migliorini and all pooled data from 68 studies, in which 7413 procedures were analyzed; they showed that longer intraoperative tourniquet use (from incision to wound clousure) is associated with shorter surgery duration, lower intraoperative blood loss, lower hemoglobin drops, and a lower transfusion rate [22].

The results of our study suggest that TXA can have a clinically valuable role in reducing intraoperative bleeding by counteracting the effects of deflation of PT on the fibrinolysis process, especially when the surgical team uses different techniques in PT placement (from skin incision to cement hardening, only during the cementation phase, or from cement hardening to the end of the procedure).

#### 4.1.2. TXA Method of Administration

There is still an ongoing debate regarding the use of TXA in TKA, concerning the route, dosage, and combination of both. In the AAKS guidelines, it is stated that there is no clearly superior method or combination of methods for the administration of TXA. All methods of administration effectively demonstrate equivalent efficacy in reducing calculated blood loss and the risk of transfusion during the primary TJA [11].

In our study, we performed elective, primary TKAs using PT and postoperative drainage in a patient population that had a low risk for thromboembolic complications. Based on the literature data and our institutional practices in TKA, we decided to administer two intraoperative intravenous dosages in the low to medium range [7,23,24]. This protocol was safe, simple, and anesthesiologist-led.

Clearly, there is no one method of administration to fit all clinical situations. Some methods of TXA administration, such as topical/intra-articular or the latest periarticular administration, are surgeon-led [8,25]. All these methods have their own potential benefits and adverse effects [26]. Following the conclusion of this study, our study protocol became the local institutional protocol in TKA for low-risk patients.

#### 4.1.3. Postoperative and Perioperative Blood Loss

Data from the current literature indicate that total postoperative blood loss in unilateral TKA varies significantly, ranging from 482 to 2875 mL [10]. Such blood loss can be clinically significant and lead to complications, hemodynamic instability, and organ and tissue ischemia. All these effects can be even more pronounced in a geriatric population with diminished physiological reserve [27].

In our study, in the control group (without use of TXA), total postoperative blood loss ranged from 100 to 1350 mL, with a median of 405 mL, and perioperatively from 450 to 1700 mL, with a median of 800 mL (Appendix A). 

Intravenous TXA significantly reduced total postoperative and perioperative blood loss in our study, which ranged from 50 to 850 mL with a median of 300 mL, and perioperatively from 150 to 850 mL with a median of 375 mL (Appendix A).

In two meta-analyses, the mean difference in total blood loss was 406.69 mL [4] and 454.29 mL [10], respectively.

Upon examining the longitudinal dynamics in both groups, only the control group exhibited a statistically significant difference across measured (6, 12, and 24 h) time periods. This indicates that the control group, which was given a placebo, had postoperative blood loss according to the typical dynamic, i.e., less blood loss over time (peak of fibrinolysis after 6 h and maintained up to 18 h) [21]. The TXA group had steady, small postoperative blood loss at the same time points, with no statistical difference (positive effect of TXA on the postoperative fibrinolisis process).

If we examine the dynamics between groups, there is a statistical difference in postoperative blood loss after 6 h. This may also be explained by the fact that fibrinolysis peaks six hours following TKA and that the administration of TXA in the TXA group attenuated this peak, explaining the variation in blood loss [21,28].

### 4.2. Secondary Outcomes

#### 4.2.1. Transfusion Rate

Patient blood management (PBM) is an important part of the enhanced recovery program in major orthopedic surgery [14,15]. The secondary aim of this study was to examine the specific effects of TXA on our institutional PBM in terms of optimizing or reducing the use of allogenic blood.

PBM represents a fine balance among procedures that optimize patient Hgb content preoperatively, minimize blood loss during and after surgery, and ensure the safe and rational use of blood and blood products [15]. 

Binary logistic regression analysis in our study revealed a relationship between preoperative Hgb and postoperative transfusion requirement, although preoperative Hgb levels did not differ between the groups and preoperative anemia was an exclusion criterion.

A possible explanation for these results lies in the hemoglobin content preoperatively. While the minimal requirement for Hgb was 13 g/dL for men and 12 g/dL for women, some patients had preoperative Hgb values that were on the upper end, meaning that their preoperative Hgb content was higher. These patients most likely had a higher margin for perioperative blood loss (and a lower risk of transfusion) than patients whose preoperative Hgb values were closer to the lower end of normal.

The surgical team performed a sub-vastus technique, and we used PT and postoperative elastic bandages in both groups. At approximately the same time, other PBM elements were institutionally developed, such as optimization of preoperative anemia with IV iron supplementation and autologous blood collection.

Our research showed that only 10.42% of patients in the TXA group received allogenic blood, compared to 45.83% in the control group. The difference was statistically significant. These findings are in correlation with the literature data. In their meta-analysis, Fillingham and coauthors analyzed 67 RCTs and showed that intravenous TXA administered as a single dose, either before or after incision, reduced the risk of transfusion by either 81% or 55% compared with a placebo. When multiple doses of intravenous TXA were administered, as in our study, the observed reduction in transfusion was 75% compared with a placebo [11,18]. Our orthopedic center performs about 480 primary TKAs a year; therefore, utilizing this TXA methodology could result in a large rationalization of blood product use. The economic implications of this reduction in transfusion rates could be substantial. It is challenging to calculate the actual cost of a blood unit. But according to current research, it is only about USD 200, not taking into consideration labor-based and other associated costs of transfusion [29]. Although economic elaboration was not performed in this study, it is safe to assume there would be significant reductions in transfusion costs based on these findings.

The transfusion rate in TKA can vary greatly, from 65.9% [30] and 31% [15] to only 2.8% in Canadian centers [31]. It primarily depends on how well-established PBM is in a particular center [32].

The transfusion trigger in our study was 9 g/dL. This transfusion trigger was proposed by our institutional protocol. This transfusion trigger is a more “liberal” (10 g/dL) protocol, according to the literature [33]. Even though a more restrictive protocol (7 g/dL) will reduce complications related directly to transfusions, in orthopeadic surgery, other patient outcomes, including length of stay, postoperative complication rate, and mortality rate, are unchanged [34].

#### 4.2.2. Postoperative Recovery

Improving the quality of life, advancing function, and relieving pain are the ultimate goals of TKA surgery. In their consensus statement for perioperative care in total hip replacement and total knee replacement surgery, the Enhanced Recovery After Surgery Society strongly recommends, with a high level of evidence, that tranexamic acid should be used as a part of the ERAS protocol in TJA surgeries [14]. 

The surgical team that carried out our study put in place an early rehabilitation regimen with the majority of postoperative elements of the ERAS protocol, consisting of a multimodal program of postoperative analgesia, with restricted usage of opioids only on the zero postoperative day, early initiation of oral ingestion of fluids and food, and early verticalization and ambulation. Both groups followed the same program, led by a ward physiotherapist. In the study group, more than 80% of patients had their first full meal on day zero, compared to 60% of patients in the control group. There was a statistically significant difference between the groups (Appendix A). Returning to normal food intake is considered an essential component of ERAS protocols [14]. No research so far has examined the direct impact of nutrition on the discharge criteria or accelerated recovery, but in the authors opinion, patients who were able to eat their first meal on day zero postoperatively were assumed not to have nausea, vomiting, severe pain, or other symptoms associated with their condition.

In our research, the TXA group experienced a faster overall postoperative recovery in terms of verticalization and ambulation. On postoperative day zero, in the TXA group, 58% of patients were standing and 50% were walking with a walking frame, in contrast to the control group, where only 12.5% were standing and 4.2% were walking with a walking frame (Appendix A).

The current literature shows that a patient’s stay in the hospital after TKA surgery varies [35]. Nowadays, the majority of patients after TKA are routinely discharged on the first or second postoperative day [14]. In selected patients, 5% of TKAs are performed as outpatient surgery (<1 postoperative day) [36].

In our study, the length of the patients’ hospital stay ranged between 3 and 15 days, while half of the patients in both groups were discharged after 3 to 5 days. There was no statistically significant difference in the length of stay between the groups.

Even though it is noticeably better than in the control group, postoperative recovery in our patients could be even further improved.

##### Study Limitations

However, there were several limitations to this study. Firstly, this was a single-center study with a limited number of patients. Secondly, this study was underpowered to draw conclusions about safety, as thromboembolic complications were monitored using only clinical parameters, so some cases may have gone undetected. The postoperative follow-up was short, up to 19 days, but thromboembolic events occurring within this period could not be attributed to the TXA effects. Another limitation is the gravimetric method, which is a form of estimation of blood loss and is therefore subject to some degree of imprecision. There is currently no clinically practical method for measuring intraoperative blood loss that is completely objective.

## 5. Conclusions

To summarize, TXA is an effective drug for reducing the incidence of perioperative bleeding and lowering transfusion rates, with an indirect positive impact on postoperative recovery in patients undergoing primary TKA.

## Figures and Tables

**Figure 1 medicina-60-01183-f001:**
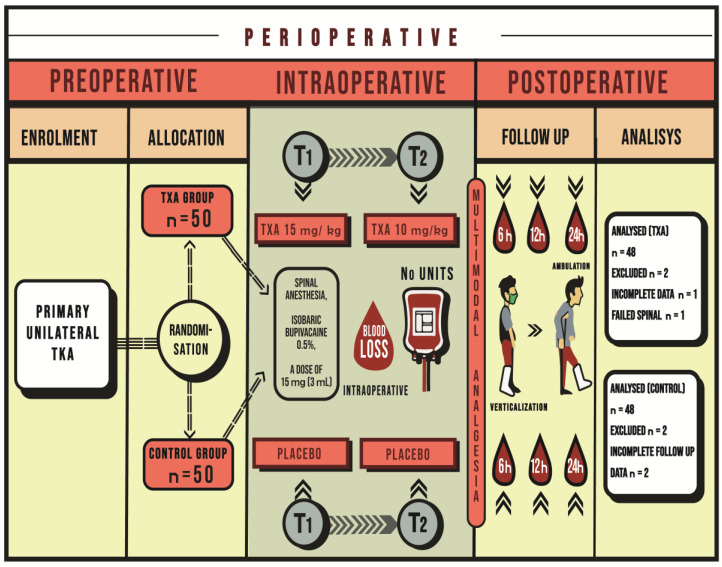
Graphic representation of the study design.

**Figure 2 medicina-60-01183-f002:**
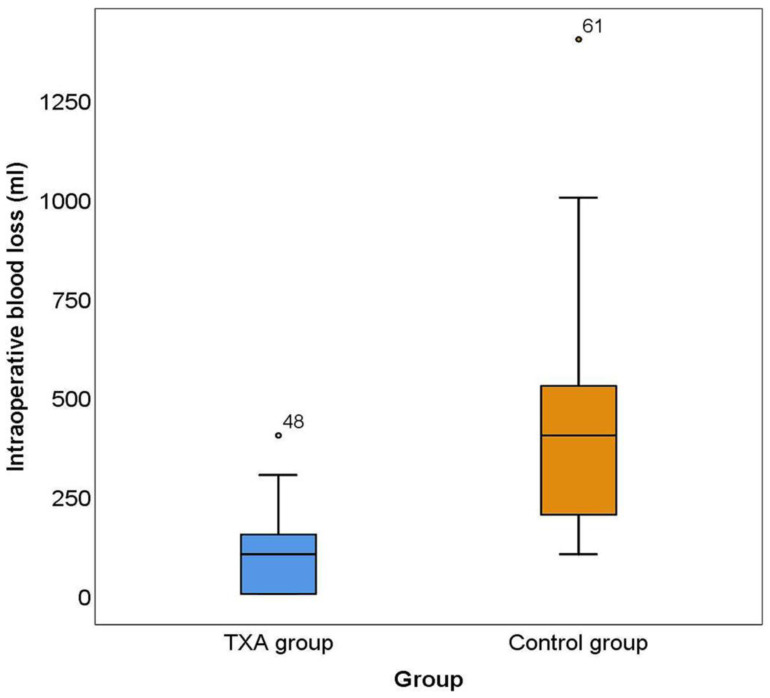
The box and whisker plot for intraoperative bleeding (mL). The box and whisker plot shows the median, minimum and maximum, interquartile range (IQR), and outliers (black circles), for intraoperative blood loss in both groups. There was a significantly higher intraoperative blood loss in the control group compared to the TXA group (Z = −6.931; *p* < 0.001).

**Figure 3 medicina-60-01183-f003:**
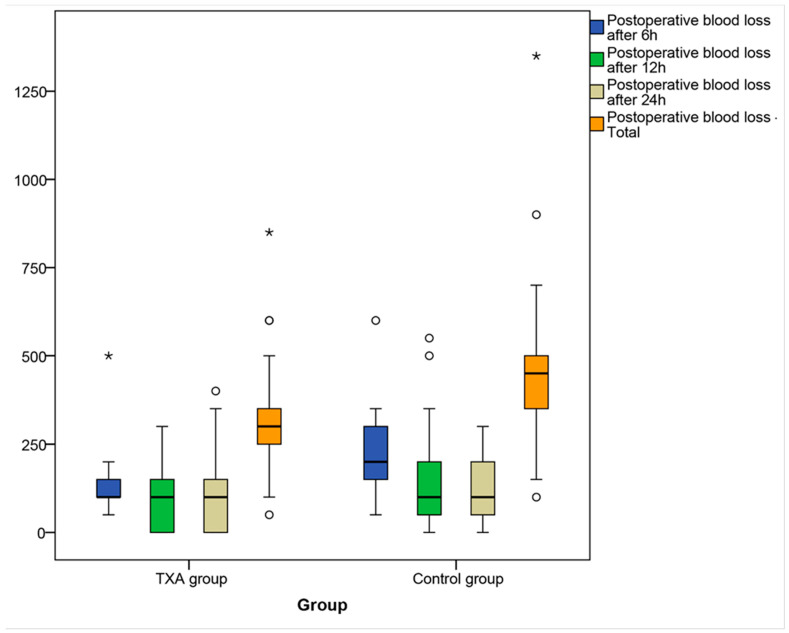
The box and whisker plot showing postoperative blood loss after 6 h, 12 h, and 24 h and total postoperative blood loss. The box and whisker plot shows the median, IQR interquartile range, minimum and maximum, outliers (black circles), and extreme values (asterisk) of postoperative blood loss after 6 h, 12 h, and 24 h and total postoperative blood loss. There was a statistically significant difference between groups after 6 h (Z = −4.511, *p* < 0.001) and in total postoperative blood loss (Z = −4.319, *p* < 0.001).

**Table 1 medicina-60-01183-t001:** Patient characteristics and TKA surgery aspects.

Items	TXA Group (n = 48)		Control Group (n = 48)		*p*
	N	% or SD	N	% or SD	
Age (years)	66.10	6.442	65.90	8.296	^†^ ns
Gender (Male/female)	9/39	18.8/81.3	11/37	22.9/77.1	^‡^ ns
BMI (kg/m^2^)	30.37	5.320	31.49	6.917	^†^ ns
ASA status					
2 group	32	66.7	24	50.0	^‡^ ns
3 group	16	33.3	24	50.0	
Hemoglobin (g/L)	137.25	13.209	135.02	12.248	^†^ ns
Operation duration (minutes)	67.92	9.556	75.08	19.418	^† *^
Tourniquet duration (minutes)	43.79	6.575	41.25	16.341	^†^ ns
Pressure in PT (mmHg)	246.04	23.856	260.42	33.196	^† *^

TKA—total knee arthroplasty; TXA—tranexamic acid; n—number; %—percent; *p*—*p* value; BMI—body mass index; ASA—American Society of Anesthesiologists; PT—pneumatic tourniquet, ^‡^ χ2 test; SD—standard deviation; ^†^ *t* test; ns—nonsignificant; * *p* < 0.05.

**Table 2 medicina-60-01183-t002:** Transfusion requirements.

Allogeneic Blood (Units)	TXA Group		Control Group		Total	
	n	%	n	%	n	%
0	43	89.6	26	54.2	69	71.9
1	5	10.4	18	37.5	24	24.0
2	0	0.0	3	6.3	3	3.1
3	0	0.0	1	2.1	1	1.0

TXA—tranexamic acid; n—number of patients; %—percent.

**Table 3 medicina-60-01183-t003:** Predictors of transfusion probability.

Variable	β	OR	95% CI	*p*
Age	−0.556	0.574	0.13–2.47	0.455
Gender	0.094	1.099	0.14–8.74	0.929
BMI (kg/m^2^)	0.671	1.957	0.41–9.30	0.399
Control group	−2.113	0.121	0.02–0.65	0.014
Preoperative hemoglobin (g/L)	0.008	1.008	1.00–1.01	0.005
Intraoperative blood loss (mL)	−0.088	0.915	0.85–0.98	0.015

β—beta value; OR—odds ratio; CI—confidence interval; *p*—*p* value; BMI—body mass index.

## Data Availability

Individual participant data will be available that underlie the results reported in this research (text, tables, figures), study protocol, and statistical analysis plan. Data will be available immediately following publication, up to 5 years after publication, to researchers who provide a methodologically sound proposal for any purpose. Proposals should be directed to principal investigator Goradana Jovanovic at the email address gordana.jovanovic@mf.uns.ac.rs, and data will be delivered accordingly.

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
