# Peer review of "The Effect of Intravenous Tranexamic Acid on Perioperative Blood Loss, Transfusion Requirements, Verticalization, and Ambulation in Total Knee Arthroplasty: A Randomized Double-Blind Study"

_medicina, 2024, doi:10.3390/medicina60071183_

Round 1

Reviewer 1 Report

Comments and Suggestions for Authors

«Numerical data were presented through mean arithmetic values and standard deviation, while comparisons of the examined groups were performed using the t-test and the Mann-Whitney U test. Pearson’s 2 test was used to test the difference in frequency (distribution) of the observed parameters». 

  • Did the authors perform distribution analysis? If the distribution is not normal, data should be presented as median and interquartile range
  • A statistical analysis is lack of information about qualitative data presentation
  • Pearson’s 2 test is not suitable for non-parametric data
  • Why the hemoglobin level at which allogeneic blood transfusion was initiated in this study was 9 g/dl? 
  • The figure’s footnotes are not correct: it’d be full explanation of bold line, boxes, whiskers and circles
  • There is no information about infusion therapy during surgery and up to 24 h in ICU. There is crucial to describe and discuss fluid balance.
  • There should be more fresh references, and the reference to the most influencing trial CRASH-2
Comments on the Quality of English Language

Minor editing of English language required

Author Response

Re

Response to Reviewer 1 Comments

1. Summary

Dear reviewer, thank you very much for taking the time to review this manuscript, and for the prompt review process. We find your comments both helpful and constructive. Please find the detailed responses below and the corresponding revisions/corrections highlighted/in track changes in the re-submitted manuscript.

2. Questions for General Evaluation

Reviewer’s Evaluation

Response and Revisions

Does the introduction provide sufficient background and include all relevant references?

Yes/Can be improved/Must be improved/Not applicable

[Please give your response if necessary. Or you can also give your corresponding response in the point-by-point response letter. The same as below]

Are all the cited references relevant to the research?

Yes/Can be improved/Must be improved/Not applicable

Is the research design appropriate?

Yes/Can be improved/Must be improved/Not applicable

Are the methods adequately described?

Yes/Can be improved/Must be improved/Not applicable

Are the results clearly presented?

Yes/Can be improved/Must be improved/Not applicable

Are the conclusions supported by the results?

Yes/Can be improved/Must be improved/Not applicable

3. Point-by-point response to Comments and Suggestions for Authors

Comments 1: Did the authors perform distribution analysis? If the distribution is not normal, data should be presented as median and interquartile range

Response 1: Yes, we performed distribution analysis. The normality of the data was assessed using the skewness, kurtosis, Kolmogorov–Smirnov tests, and Shapiro –Wilik test.

We have included two additional tables in the revised manuscript, S1 and S2, in the supplementary section with the data.

Table S1. Descriptive data on interoperative, postoperative and perioperative blood loss (ml) and results of group comparations (Man-Witney Test)

Table S2. Descriptive data about postoperative recovery and distribution analysis (Chi-Square (c2) Test)

Comments 2: A statistical analysis is lack of information about qualitative data presentation

Response 2: We are very sorry but seems that we do not fully understand this remark. Could you please rephrase this question?

Comments 3: Pearson’s 2 test is not suitable for non-parametric data

Response 3: We are very sorry but seems that we do not fully understand this remark. Could you please rephrase this question?

Comments 4. Why the hemoglobin level at which allogeneic blood transfusion was initiated in this study was 9 g/dl? 

Response 4: Thank you for this remark. In our country, there are no national recommendations for transfusion (and transfusion triggers in surgery). The transfusion trigger used in the study was proposed by our institutional protocol at the time this study was conducted, which is why we used this particular transfusion trigger.

We believe that the answer to your question is very adequately adressed in the following paragraph: “There exist two main schools of thought in relation to perioperative blood transfusion - transfusing at a haemoglobin level of 70 g/L or below (termed 'restrictive protocol' in the literature), versus transfusing at a haemoglobin threshold of 100 g/L (which the literature terms 'liberal'). Few reviews of the literature exist comparing the impact of different transfusion strategies on the postoperative outcomes of these orthopaedic patients. This review analyses the 11 currently available studies on transfusion protocols in orthopaedics and associated findings related to patient outcomes. The literature showed no clear consensus on whether one transfusion protocol is superior in the orthopaedic patient. There was strong evidence for reduced transfusion rates among groups receiving transfusion at <70 g/L, and hence a reduction in complication directly related to transfusions. Despite this, other measured patient outcomes, including length of stay, postoperative complication rate, and mortality rate, were unchanged between the groups. Some evidence exists that aHb threshold of 100 g/L results in earlier postoperative ambulation in hip surgery patients.” (Pietris J. The effect of perioperative blood transfusion thresholds on patient outcomes in orthopaedic surgery: a literature review. ANZ J Surg.2022;92(4):661–665.)

Comments 5. The figure’s footnotes are not correct: it’d be full explanation of bold line, boxes, whiskers and circles,

Response 5: We agree that figure footnotes should be presented more systematically. We have changed the figure footnotes, and all the comments about figures 2 and 3 were put in the text of the revised manuscript

Comments 6 There is no information about infusion therapy during surgery and up to 24 h in ICU. There is crucial to describe and discuss fluid balance.

Response 6: Thank you for this remark. The research itself gathered much more data than presented in this study. We collected data on intraoperatively administered crystalloid solutions, both as a total amount per patient and as an average amount per patient in milliliters. However, fluid balance for individual patients was not calculated.

In the postoperative period, patients were on the ward and began oral ingestion of fluids as soon as the residual effects of spinal anesthesia were completely gone (typically after a couple of hours), with intravenous infusions administered only for drug administration.

We will include this data in the revised manuscript.

“Intraoperatively, the average amount of administered crystalloid solutions (Ringer lactate) per patient was 1844.68 ml in the TXA group and 1597.92 ml in the control group, reaching a statistically significant difference.”

Comments 7. There should be more fresh references, and the reference to the most influencing trial CRASH-2

Response 7: Thank you for this remark. We have updated the reference list in the revised manuscript to align with the changes made in the introduction and discussion sections.

4. Response to Comments on the Quality of English Language

Point 1: Minor editing of English language required

Response 1:  Revised manuscript is checked by professor of Engulish language.

5. Additional clarifications

[We provided detailed revision of the "Material and Methods" section too.

Here, mention any other clarifications you would like to provide to the journal editor/reviewer.]

sponse to Reviewer 2 Comments

1. Summary

Dear reviewer, thank you very much for taking the time to review this manuscript, and for the prompt review process. We find your comments both helpful and constructive. Please find the detailed responses below and the corresponding revisions/corrections highlighted/in track changes in the re-submitted manuscript.

2. Questions for General Evaluation

Reviewer’s Evaluation

Response and Revisions

Does the introduction provide sufficient background and include all relevant references?

Yes/Can be improved/Must be improved/Not applicable

[Please give your response if necessary. Or you can also give your corresponding response in the point-by-point response letter. The same as below]

Are all the cited references relevant to the research?

Yes/Can be improved/Must be improved/Not applicable

Is the research design appropriate?

Yes/Can be improved/Must be improved/Not applicable

Are the methods adequately described?

Yes/Can be improved/Must be improved/Not applicable

Are the results clearly presented?

Yes/Can be improved/Must be improved/Not applicable

Are the conclusions supported by the results?

Yes/Can be improved/Must be improved/Not applicable

3. Point-by-point response to Comments and Suggestions for Authors

Comments 1: Did the authors perform distribution analysis? If the distribution is not normal, data should be presented as median and interquartile range

Response 1: Yes, we performed distribution analysis. The normality of the data was assessed using the skewness, kurtosis, Kolmogorov–Smirnov tests, and Shapiro –Wilik test.

We have included two additional tables in the revised manuscript, S1 and S2, in the supplementary section with the data.

Table S1. Descriptive data on interoperative, postoperative and perioperative blood loss (ml) and results of group comparations (Man-Witney Test)

Table S2. Descriptive data about postoperative recovery and distribution analysis (Chi-Square (c2) Test)

Comments 2: A statistical analysis is lack of information about qualitative data presentation

Response 2: We are very sorry but seems that we do not fully understand this remark. Could you please rephrase this question?

Comments 3: Pearson’s 2 test is not suitable for non-parametric data

Response 3: We are very sorry but seems that we do not fully understand this remark. Could you please rephrase this question?

Comments 4. Why the hemoglobin level at which allogeneic blood transfusion was initiated in this study was 9 g/dl? 

Response 4: Than k you for this remark. In our country, there are no national recommendations for transfusion (and transfusion triggers in surgery). The transfusion trigger used in the study was proposed by our institutional protocol at the time this study was conducted, which is why we used this particular transfusion trigger.

We believe that the answer to your question is very adequately adressed in the following paragraph: “There exist two main schools of thought in relation to perioperative blood transfusion - transfusing at a haemoglobin level of 70 g/L or below (termed 'restrictive protocol' in the literature), versus transfusing at a haemoglobin threshold of 100 g/L (which the literature terms 'liberal'). Few reviews of the literature exist comparing the impact of different transfusion strategies on the postoperative outcomes of these orthopaedic patients. This review analyses the 11 currently available studies on transfusion protocols in orthopaedics and associated findings related to patient outcomes. The literature showed no clear consensus on whether one transfusion protocol is superior in the orthopaedic patient. There was strong evidence for reduced transfusion rates among groups receiving transfusion at <70 g/L, and hence a reduction in complication directly related to transfusions. Despite this, other measured patient outcomes, including length of stay, postoperative complication rate, and mortality rate, were unchanged between the groups. Some evidence exists that aHb threshold of 100 g/L results in earlier postoperative ambulation in hip surgery patients.” (Pietris J. The effect of perioperative blood transfusion thresholds on patient outcomes in orthopaedic surgery: a literature review. ANZ J Surg.2022;92(4):661–665.)

Comments 5. The figure’s footnotes are not correct: it’d be full explanation of bold line, boxes, whiskers and circles,

Response 5: We agree that figure footnotes should be presented more systematically. We have changed the figure footnotes, and all the comments about figures 2 and 3 were put in the text of the revised manuscript

Comments 6 There is no information about infusion therapy during surgery and up to 24 h in ICU. There is crucial to describe and discuss fluid balance.

Response 6: Thank you for this remark. The research itself gathered much more data than presented in this study. We collected data on intraoperatively administered crystalloid solutions, both as a total amount per patient and as an average amount per patient in milliliters. However, fluid balance for individual patients was not calculated.

In the postoperative period, patients were on the ward and began oral ingestion of fluids as soon as the residual effects of spinal anesthesia were completely gone (typically after a couple of hours), with intravenous infusions administered only for drug administration.

We will include this data in the revised manuscript.

“Intraoperatively, the average amount of administered crystalloid solutions (Ringer lactate) per patient was 1844.68 ml in the TXA group and 1597.92 ml in the control group, reaching a statistically significant difference.”

Comments 7. There should be more fresh references, and the reference to the most influencing trial CRASH-2

Response 7: Thank you for this remark. We have updated the reference list in the revised manuscript to align with the changes made in the introduction and discussion sections.

4. Response to Comments on the Quality of English Language

Point 1: Minor editing of English language required

Response 1:  Revised manuscript is checked by professor of Engulish language.

5. Additional clarifications

[Here, mention any other clarifications you would like to provide to the journal editor/reviewer.]

Reviewer 2 Report

Comments and Suggestions for Authors

Thank you for your effort on the study and the manuscript.

My comments:

-The introduction section should be rewritten. Instead of giving historical and chemical features of the TXA, it could be better to provide information about the TXA used in previous studies on bleeding during arthroplasty cases.

-The material methods section: In the introduction section you mentioned 10 mg/kg is enough for sufficient plasma concentration; so why did you give patients  15 mg/kg? 

For intraoperative blood loss, did you compare groups after tourniquet release?

In the results section: What was the reason for the longer surgical time in the control group while the tourniquet time is shorter?

Why was PT pressure higher in the control group?

The discussion section: 

This section is not well-written. In this section, you should discuss your results with the literature, but there is nearly no discussion in the current form.

You should add limitations of your study. 

Author Response

Response to Reviewer 2 Comments

1. Summary

Dear reviewer, thank you very much for taking the time to review this manuscript, and for the prompt review process. We find your comments both helpful and constructive. Please find the detailed responses below and the corresponding revisions/corrections highlighted/in track changes in the re-submitted manuscript.

2. Questions for General Evaluation

Reviewer’s Evaluation

Response and Revisions

Does the introduction provide sufficient background and include all relevant references?

Yes/Can be improved/Must be improved/Not applicable

[Please give your response if necessary. Or you can also give your corresponding response in the point-by-point response letter. The same as below]

Are all the cited references relevant to the research?

Yes/Can be improved/Must be improved/Not applicable

Is the research design appropriate?

Yes/Can be improved/Must be improved/Not applicable

Are the methods adequately described?

Yes/Can be improved/Must be improved/Not applicable

Are the results clearly presented?

Yes/Can be improved/Must be improved/Not applicable

Are the conclusions supported by the results?

Yes/Can be improved/Must be improved/Not applicable

3. Point-by-point response to Comments and Suggestions for Authors

-Comments 1:

The introduction section should be rewritten. Instead of giving historical and chemical features of the TXA, it could be better to provide information about the TXA used in previous studies on bleeding during arthroplasty cases.

Response 1: Thank you for pointing this out. We comply with this comment. The revised manuscript will have rewritten introduction.

Comments 2: The material methods section: In the introduction section you mentioned 10 mg/kg is enough for sufficient plasma concentration; so why did you give patients  15 mg/kg? 

Response 2: The full statement about concentration declares :”Plasma concentration of 10 ng/mL and an 80% reduction in the activity of plasminogen activator for adequate suppression of fibrinolysis in tissues. An intravenous dose of tranexamic acid of 10 mg/kg maintains such a plasma concentration for approximately 3 hours. ( Nilsson IM. Clinical pharmacology of aminocaproic and tranexamic acids. J ClinPatholSuppl (R CollPathol).1980;14:41–47).

There is still no clinical consensus on the optimal dosing of TXA in TKA. Doses can be high (> 20mg/kg) or low (<20 mg/kg). We chose a protocol of two intraoperative intravenous doses, the first dose being 15mg/kg and the next dose being 10 mg /kg, because it was simple, still within the range of low dosages, cost-effective, and applicable in our clinical situation. We opted for a slightly higher first dose because of the use of a pneumatic tourniquet and to prolong the effect into the postoperative period. After the study, this protocol became our local institutional protocol for TKA.

Comments 3: For intraoperative blood loss, did you compare groups after tourniquet release?

Response 3: All surgeries (TKA) in this study were performed with the placement of a PT at the proximal thigh before the start of the incision, ensuring a clear and avascular operative field. At the surgeon’s discretion, the tourniquet was deflated after cementing the prosthesis. During the tourniquet time, blood loss was negligible, so all intraoperative blood loss occurred after the tourniquet was released. Therefore, we can say that intraoperative blood loss (in total) refers to the blood loss after the tourniquet release. “The average intraoperative blood loss in the study group was 100 ± 92.69 milliliters, and in the control group, it was 447 ± 299.28 milliliters (Figure 2). The results of the Mann-Whitney U test show a statistically significantly higher intraoperative bleeding in the control group compared to the study group (Z = -7.281; p <0.001.)

Comments 4. In the results section: What was the reason for the longer surgical time in the control group while the tourniquet time is shorter?

Response 4: There is no apparent clinical reason. All surgeries were performed by the same surgical team. If we analyze the difference in the duration of TT in the control group, which is 41.25 min on average vs. 43.79 min in the TXA group, it is clear that the difference between the groups is 2.54 min on average. We believe this difference is not clinically significant. The same applies to the difference in operation time, which is 7.16 min on average.

Comments 5. Why was PT pressure higher in the control group?

Response 5: There is no consensus on how to calculate safe tourniquet pressure. In our clinical practice, we calculated tourniquet pressure as a fixed amount above the systolic arterial pressure at the beginning of the procedure, before the administration of spinal anesthesia. (typically þ100 mm Hg for the upper arm and þ100–150 mm Hg for the thigh) .(Deloughry JL, Griffiths R. Arterial tourniquets. Continuing Education in Anesthesia Crit Care and Pain (Indian Edition) 2009;2:64-8.). We believe this difference is solely due to varying systolic pressures among patients in the control group.

Comments 6 The discussion section: This section is not well-written. In this section, you should discuss your results with the literature, but there is nearly no discussion in the current form.

Response 6:Thank you for pointing this out. We comply with this comment. The revised manuscript will have rewritten discussion section.

Comments 7.You should add limitations of your study. 

Response 7: Yes we agree, and we will add this text in the final manuscript:

However, there were several limitations in this study. Firstly, this was a single-center study with a limited number of patients. Secondly, the study was underpowered to draw conclusions about safety, as thromboembolic complications were monitored using clinical parameters only, thus some cases may have gone undetected. The postoperative follow-up was short, up to 19 days, but thromboembolic events occurring within this period could not be attributed to the TXA effects

4. Response to Comments on the Quality of English Language

Point 1:

Response 1:    (in red)

5. Additional clarifications

We provided detailed revision of "Material and Methods" section too.

Round 2

Reviewer 1 Report

Comments and Suggestions for Authors
  • The figure’s footnotes are not correct: it’d be explanation of whiskers - 95% CI
  •  

Author Response

Open Review (x) I would not like to sign my review report
( ) I would like to sign my review report Quality of English Language ( ) I am not qualified to assess the quality of English in this paper
( ) English very difficult to understand/incomprehensible
( ) Extensive editing of English language required
( ) Moderate editing of English language required
( ) Minor editing of English language required
(x) English language fine. No issues detected            
  Yes Can be improved Must be improved Not applicable
Does the introduction provide sufficient background and include all relevant references? (x) ( ) ( ) ( )
Is the research design appropriate? (x) ( ) ( ) ( )
Are the methods adequately described? ( ) (x) ( ) ( )
Are the results clearly presented? ( ) (x) ( ) ( )
Are the conclusions supported by the results? (x) ( ) ( ) ( )
Dear reveiwer, thank you for the prompt reply.  Comment 1: Comments and Suggestions for Authors The figure’s footnotes are not correct: it’d be explanation of whiskers - 95% CI Reply 1: Thank you for the observations. In the text explaining the Figure 2 and Figure 3 we added 95% CI for the all median values. They  are marked green in the text of revised manuscript.

Reviewer 2 Report

Comments and Suggestions for Authors

Thank you for your effort on the paper and the revision.

The current form of paper can be accepted to publicaiton.

Author Response

Comment 1. Comments and Suggestions for Authors

Thank you for your effort on the paper and the revision.

The current form of paper can be accepted to publicaiton.

Reply 1: Dear Reviewer thank you for the prompt review, your comments were both helpful and constructive, and greatly  added to the revised manuscript  quality.